# Discriminative Sounding Objects Localization via Self-supervised Audiovisual Matching

**Di Hu**[1,2]\*, **Rui Qian**[3], **Minyue Jiang**[2], **Xiao Tan**[2], **Shilei Wen**[2], **Errui Ding**[2],
**Weiyao Lin**[3], **Dejing Dou**[2]
[1]Renmin University of China, [2]Baidu Inc., [3]Shanghai Jiao Tong University
dihu@ruc.edu.cn,{qrui9911,wylin}@sjtu.edu.cn,
{jiangminyue,wenshilei,dingerrui,doudejing}@baidu.com,tanxchong@gmail.com

## Abstract

Discriminatively localizing sounding objects in cocktail-party, i.e., mixed sound scenes, is commonplace for humans, but still challenging for machines. In this paper, we propose a two-stage learning framework to perform self-supervised class-aware sounding object localization. First, we propose to learn robust object representations by aggregating the candidate sound localization results in the single source scenes. Then, class-aware object localization maps are generated in the cocktail-party scenarios by referring the pre-learned object knowledge, and the sounding objects are accordingly selected by matching audio and visual object category distributions, where the audiovisual consistency is viewed as the self-supervised signal. Experimental results in both realistic and synthesized cocktail-party videos demonstrate that our model is superior in filtering out silent objects and pointing out the location of sounding objects of different classes. Code is available at `https://github.com/DTaoo/Discriminative-Sounding-Objects-Localization`.

## 1   Introduction

Audio and visual messages are pervasive in our daily-life. Their natural correspondence provides humans with rich semantic information to achieve effective multi-modal perception and learning [28, 24, 15], e.g., when in the street, we instinctively associate the talking sound with people nearby, and the roaring sound with vehicles passing by. In view of this, we want to question that can they also facilitate machine intelligence?

To pursue the human-like audiovisual perception, the typical and challenging problem of visually sound localization is highly expected to be addressed, which aims to associate sounds with specific visual regions and rewards the visual perception ability in the absence of semantic annotations [14, 18, 3, 27, 10]. A straightforward strategy is to encourage the visual features of sound source to take higher similarity with the sound embeddings, which has shown considerable performance in the simple scenarios with single sound [21, 22, 27]. However, there are simultaneously multiple sounding objects as well as silent ones (i.e. The silent objects are considered capable of producing sound.). in our daily scenario, i.e., the cocktail-party, this simple strategy mostly fails to discriminatively localize different sound sources from mixed sound [16]. Recently, audiovisual content modeling is proposed to excavate concrete audio and visual components in the scenario for localization. Yet, due to lack of sufficient semantic annotation, existing works have to resort to extra scene prior knowledge [16, 17, 25] or construct pretext task [31, 30]. Even so, these methods cannot well deal

with such complex cocktail-party scenario, i.e., not only answering *where the sounding area is* but also answering *what the sounding area is*.

In this paper, we target to perform class-aware sounding object localization from their mixed sound, where the audiovisual scenario consists of multiple sounding objects and silent objects, as shown in Fig. 1. This interesting problem is quite challenging from two perspectives: 1) Discriminatively localizing objects belonging to different categories without resorting to semantic annotations of objects; 2) Determining whether a specific object is sounding or not, and filtering out silent ones from the corresponding mixed sound. When faced with these challenges, we want to know how do we human address them? Elman [9] stated that human could transform these seemingly unlearnable tasks into learnable by starting from a simpler initial state then building on which to develop more complicated representations of structure. Inspired by this, we propose a two-stage framework, evolving from single sound scenario to the cocktail-party case. Concretely, we first learn potential object knowledge from sound localization in single source scenario, and aggregate them into a dictionary for pursuing robust representation for each object category. By referring to the dictionary, class-aware object localization maps are accordingly proposed for meeting the sounding object selection in multi-source scenario. Then,

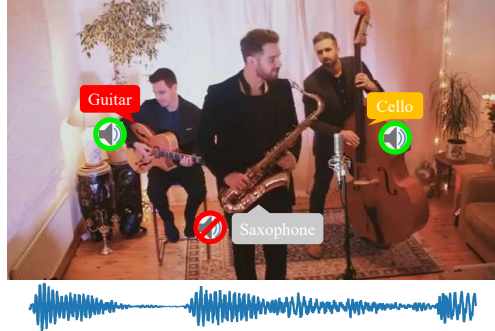

Figure 1: An example of cocktail-party scenario, which contains sounding guitar, sounding cello and silent saxophone. We aim to discriminatively localize the sounding instruments and filter out the silent ones. Video URL: https://www.youtube.com/watch?v=ebugBtNiDMI.

we reduce the sounding object localization task into a self-supervised audiovisual matching problem, where the sounding objects are selected by minimizing the category-level audio and visual distribution difference. With these evolved curriculums, we can filter out silent objects and achieve class-aware sounding object localization in a cocktail-party scenario.

To summarize, our main contributions are as follows. **First**, we introduce an interesting and challenging problem, i.e., discriminatively localizing sounding objects in the cocktail-party scenario without manual annotation for objects. **Second**, we propose a novel step-by-step learning framework, which learns robust object representations from single source localization then further expands to the sounding object localization via taking audiovisual consistency as self-supervision for category distribution matching in the cocktail-party scenario. **Third**, we synthesize some cocktail-party videos and annotate sounding object bounding boxes for the evaluation of class-aware sounding object localization. Our method shows excellent performance on both synthetic and realistic data.

## 2 Related work

**Object localization** Weakly- and self-supervised object localization expect to achieve comparable performance to the supervised ones with limited annotations. Existing weakly-supervised methods take holistic image labels as supervision, where the salient image region evaluated by recognition scores are considered as the potential object location[20, 21, 6, 32, 26, 7]. For self-supervised models, Baek et al. [5] used point symmetric transformation as self-supervision to extract class-agnostic heat maps for object localization. These methods are purely based on visual features, while we propose to employ audiovisual consistency as self-supervision to achieve class-aware object localization.

**Self-supervised audiovisual learning** The natural correspondence between sound and vision provides essential supervision for audiovisual learning [2, 3, 22, 4, 23]. In [23, 4], authors introduced to learn feature representations of one modality with the supervision from the other. In [2, 22], authors adopted clip-level audiovisual correspondence and temporal synchronization as self-supervision to correlate audiovisual content. Hu et al. [16, 17] associate latent sound-object pairs with clustered audiovisual components, but its performance greatly relies on predefined number of clusters. Alwassel et al. [1] created pseudo labels from clustering features to boost multi-modal representation learning. While in our work, we alternatively use audiovisual correspondence and pseudo labels from clustering to boost audiovisual learning and learn object representations.

**Sounding object localization in visual scenes** Recent methods for localizing sound source in visual context mainly focus on joint modeling of audio and visual modalities [3, 22, 27, 29, 16, 30, 31]. In [3, 22], authors adopted *Class Activation Map* (CAM) [32] or similar methods to measure the correspondence score between audio and visual features on each spatial grid to localize sounding objects. Senocak et al. [27] proposed an attention mechanism to capture primary areas in a semi-supervised or unsupervised setting. Tian et al. [29] leveraged audio-guided visual attention and temporal alignment to find semantic regions corresponding to sound sources. These methods tend to perform well in single source scenes, but comparatively poor for mixed sound localization. Zhao et al. [31, 30] employed a sound-based mix-then-separate framework to associate the audio and visual feature maps, where the sound source position is given by the sound energy of each pixel. Hu et al. [16] established audiovisual clustering to associate sound centers with corresponding visual sources, but it requires the prior of the number of sound sources, and the specific category of the clustering result remains unknown. In contrast, our method can discriminatively localize sounding objects in cocktail-party by employing established object dictionary to generate class-aware object localization maps, and referring to the audiovisual localization map to filter out the silent ones.

## 3 The proposed method

In this work, we aim to discriminatively localize the sounding objects from their mixed sound without the manual annotations of object category. To facilitate this novel and challenging problem, we develop a two-stage learning strategy, evolving from the localization in simple scenario with single sounding object to the complex one with multiple sounding objects, i.e., cocktail-party. Such curriculum learning perspective is based on the findings that existing audiovisual models [3, 16, 27] are capable of predicting reasonable localization map of sounding object in simple scenario, which is considered to provide effective knowledge reference for candidate visual localization of different objects in the cocktail-party scenario.

Specifically, for a given set of audiovisual pair with arbitrary number of sounding objects, $\mathcal{X} = \{(a_i, v_i)| i = 1, 2, ..., N\}$, we first divide it into one simple set whose scenario only contains single sounding object, $\mathcal{X}^s = \{(a_i^s, v_i^s)| i = 1, 2, ..., N^s\}$, and one complex set, where each audiovisual pair consists of several sounding objects, $\mathcal{X}^c = \{(a_i^c, v_i^c)| i = 1, 2, ..., N^c\}$, where $\mathcal{X} = \mathcal{X}^s \cup \mathcal{X}^c$ and $\mathcal{X}^s \cap \mathcal{X}^c = \emptyset$. In the first stage, we propose to learn potential visual representation of sounding object from their localization map in the simple scenario $\mathcal{X}^s$, with which we build a representation dictionary of objects as a kind of visual object knowledge reference. In the second stage, by referring to the learned representation dictionary, we step forward to discriminatively localize multiple sounding objects in the complex scenario $\mathcal{X}^c$, where the category distribution of localized sounding objects are required to match the distribution of their mixed sound according to the natural audiovisual consistency [16]. In the rest sections, we detail the first and second learning stage for generalized sounding object localization.

### 3.1 Learning object representation from localization

For the simple audiovisual scenario with single sound source, $\mathcal{X}^s$, we target to visually localize the sounding object from its corresponding sound, and synchronously build a representation dictionary from the localization outcomes. The framework is shown in the left part of Fig. 2.

At the first step, given an arbitrary audiovisual pair $(a_i^s, v_i^s) \in \mathcal{X}^s$, to exactly predict the position of sounding object, we need to find which region of input image $v_i^s$ is highly correlated to the sound $a_i^s$. To this end, we feed the image into a convolution-based network (e.g., ResNet [13]) to extract spatial feature maps $f(v_i^s) \in R^{C \times H \times W}$ as the local image region descriptors, where $C$ is the channel dimension, $H$ and $W$ are the spatial size. Then, the localization network is encouraged to enhance the similarity between the image region of sounding object and corresponding sound embeddings $g(a_i^s)$ from the same video, but suppress those ones where sound and object are mismatched (from different videos), i.e., $(a_i^s, v_j^s)$, where $i \neq j$. Formally, the localization objective can be written as

$$\mathcal{L}_1 = \mathcal{L}_{bce}(y^{match}, GMP(l(g(a_i^s), f(v_j^s)))), \tag{1}$$

where the indicator $y^{match} = 1$ is the audio and image are from the same pair, i.e., $i = j$, otherwise $y^{match} = 0$, and $\mathcal{L}_{bce}$ is the binary cross-entropy loss. $l(g(a_i^s), f(v_j^s))$ is the audiovisual localization function, achieved by computing the cosine similarity of audio and visual feature representation[2].

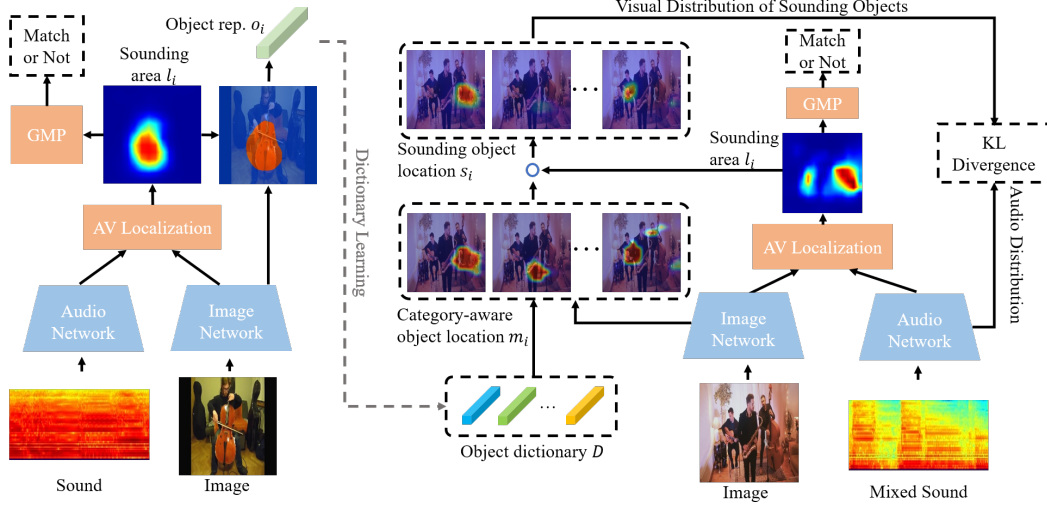

Figure 2: An overview of the proposed framework. First, we use audiovisual correspondence as the self-supervision to localize sounding area and learn object representation (left). Then, we employ built object dictionary to generate class-aware localization maps and refer to inferred sounding area to eliminate silent objects. In that way, we reduce the localization task into a distribution matching problem, where we use KL divergence to minimize audiovisual distribution difference (right).

Similar to [3], *Global Max Pooling* (GMP) is used to aggregate the localization map to match the scene-level supervision. As there is no extra semantic annotation employed, the localization model is fully optimized in a self-supervised fashion.

As the localization map could provide effective reference of object position, it helps to reduce the disturbance of complex background and boosts the visual perception performance of object appearance. To supply better visual object reference for the multi-source localization in the second stage, we utilize these localization outcomes to learn a kind of representation dictionary $D$ for different object categories. First, we propose to binarize the localization map $l_i$ of the $i-$th audiovisual pair into a mask $m_i \in \{0,1\}^{H \times W}$. As there should be only one sounding object in the simple scenario $\mathcal{X}^s$, $m_i$ should be a single-object-awareness mask indicator. Hence, we can extract potential object representation $o_i \in R^C$ over the masked visual features $f(v_i^s)$, i.e.,

$$o_i = GAP(f(v_i^s) \circ m_i), \tag{2}$$

where $GAP$ is the Global Average Pooling operation and $\circ$ is the Hadamard product. These object representations $\mathcal{O} = \{o_1, o_2, ..., o_{N^s}\}$ are extracted from the coarse localization results, which makes it difficult to provide robust expression of object characters. To facilitate such progress, we target to learn high-quality object indicators with these candidate representations in a dictionary learning fashion. Specifically, we propose to jointly learn a $K \times C$ dictionary $D$ and assignment $y_i$ of each object representation $o_i$, where each key $d^k \in R^{1 \times C}$ is identified as the representative object character in the $k-$th category. As K-means can be viewed as an efficient way of constructing representation dictionary [8], in our case we aim to minimize the following problem,

$$\mathcal{L}(D, y_i) = \sum_{i=1}^{N^s} \min_{y_i} ||o_i - D^T \cdot y_i||_2^2 \quad s.t. \ y_i \in \{0,1\}^K, \sum y_i = 1, \tag{3}$$

where $K$ is the number of object category. Solving this problem provides a dictionary $D^*$ and a set of category assignments $\{y_i^* | i = 1, 2, ...N^s\}$, where the former one is used for potential object detection in the second stage and the latter can be viewed as pseudo labels indicating different object categories. Recall that object localization could benefit from generalized categorization [21, 32], we therefore choose to alternately optimize the model w.r.t. the localization objective using Eq. 1 and the object classification objective with generated pseudo labels, which could substantially improve the localization performance.

## 3.2 Discriminative sounding object localization

To discriminatively localize different sounding objects from their mixed sound, we propose to localize all the emerged objects in the image first, among which the sounding ones are causally selected based on whether they appear in the sounding area and required to match the category distribution of corresponding audio messages, as shown in the right part of Fig. 2.

Let $(a_i^c, v_i^c) \in \mathcal{X}^c$ denote the $i-$th audiovisual message that consists of multiple sounding objects. By referring to the learned representation dictionary of objects $D^*$, the location of emerged objects is indicated by computing the following inner-product similarity between each location of visual feature map $f(v_i^c) \in R^{C \times H \times W}$ and each representation key $d^k \in R^{1 \times C}$ within $D^*$,

$$m_i^k = d^k \cdot f(v_i^c), \tag{4}$$

where $m_i^k$ is the predicted object location area of the $k-$th category in the $i-$th visual scenario. If the scenario does not involve the object belonging to the $k-$th category, the corresponding localization map $m_i^k$ tends to remain low response (similarity). At this point, we can obtain $K$ localization maps, indicating the location of different categories of objects.

As stated in the beginning, the cocktail-party scenario may consist of multiple sounding objects and silent objects. To localize the sounding objects as well as eliminate the silent ones, the sounding area $l_i$ that is highly related to the input mixed sound is regarded as a kind of sounding object filter, which is formulated as

$$s_i^k = m_i^k \circ l_i. \tag{5}$$

$s_i^k$ is deemed as the location of sounding object of the $k-$th category. Intuitively, if the $k-$th object does not produce any sound even if it visually appears in the image, there will be no sounding areas reflected in $s_i^k$. Hence, the category distribution of sounding objects for $v_i^c$ can be written as

$$p_{v_i}^{so} = softmax([GAP(s_i^1), GAP(s_i^2), ..., GAP(s_i^K)]). \tag{6}$$

As discussed in recent works [16], the natural synchronization between vision and sound provides the self-supervised consistency in terms of sounding object category distribution. In other words, the sound character and the visual appearance of the same sounding object are corresponding in taxonomy, such as barking and dog, meow and cat. Hence, we propose to train the model to discriminatively localize the sounding objects by solving the following problem,

$$\mathcal{L}_c = \mathcal{D}_{KL}(p_{v_i}^{so} || p_{a_i}^{so}), \tag{7}$$

where $p_{a_i}^{so}$ is the category distribution of sound $a_i$, predicted by a well-trained audio event network[3], and $\mathcal{D}_{KL}$ is the Kullback–Leibler divergence.

Overall, the second stage consists of two learning objective, one is the category-agnostic sounding area detection and the other one is class-aware sounding object localization, i.e.,

$$\mathcal{L}_2 = \mathcal{L}_c + \lambda \cdot \mathcal{L}_1, \tag{8}$$

where $\lambda$ is the hype-parameter balancing the importance of both objective. By solving the problem in Eq. 8, the location of sounding objects are discriminatively revealed in the category-specific maps $\{s_i^1, s_i^2, ..., s_i^K\}$. Finally, softmax regression is performed across these class-aware maps on each location for better visualization.

# 4 Experiments

## 4.1 Datasets and annotation

**MUSIC** MUSIC dataset [31] contains 685 untrimmed videos, 536 solo and 149 duet, covering 11 classes of musical instruments. To better evaluate sound localization results in diverse scenes, we use the first five/two videos of each instrument category in solo/duet for testing, and use the rest for training. Besides, we use one half of solo training data for the first-stage training, and employ the other half to generate synthetic data for the second-stage learning. Note that, some videos are now not available on YouTube, we finally get 489 solo and 141 duet videos.

Table 1: Localization results on MUSIC-solo and AudioSet-instrument-solo.

(a) Results on MUSIC-solo.

| Methods | IoU@0.5 | AUC |
|---|---|---|
| Sound-of-pixel [31] | 40.5 | 43.3 |
| Object-that-sound [3] | 26.1 | 35.8 |
| Attention [27] | 37.2 | 38.7 |
| DMC [16] | 29.1 | 38.0 |
| Ours | **51.4** | **43.6** |

(b) Results on AudioSet-instrument-solo.

| Methods | IoU@0.5 | AUC |
|---|---|---|
| Sound-of-pixel [31] | 38.2 | 40.6 |
| Object-that-sound [3] | 32.7 | 39.5 |
| Attention [27] | 36.5 | 39.5 |
| DMC [16] | 32.8 | 38.2 |
| Ours | **38.9** | **40.9** |

**MUSIC-Synthetic** The categories of instruments in duet videos of MUSIC dataset are quite unbalanced, e.g., more than 80% duet videos contain sound of guitar, which is difficult for training and brings great bias in testing. Thus, we build category-balanced multi-source videos by artificially synthesizing solo videos to facilitate our second-stage learning and evaluation. Concretely, we first randomly choose four 1-second solo audiovisual pairs of different categories, then mix random two of the four audio clips with jittering as the multi-source audio waveform, and concatenate four frames of these clips as the multi-source video frame. That is, in the synthesized audiovisual pair, there are two instruments making sound while the other two are silent. Therefore, this synthesized dataset is quite proper for the evaluation of discriminatively sounding object localization[4].

**AudioSet-instrument** AudioSet-instrument dataset is a subset of AudioSet [12], consisting of 63,989 10-second video clips covering 15 categories of instruments. Following [11], we use the videos from the "unbalanced" split for training, and those from the "balanced" for testing. We employ the solo videos with single sound source for the first-stage training and testing, and adopt those with multiple sound sources for the second-stage training and testing.

**Bounding box annotation** To quantitatively evaluate the sound localization performance, we use a well-trained Faster RCNN detector w.r.t 15 instruments [11] to generate bounding boxes on the test set. We further refine the detection results, and manually annotate whether each object is sounding or silent. Annotations are publicly available in the released code, for reproducibility.

## 4.2 Experimental settings

**Implementation details** Each video in the above datasets are equally divided into one second clips, with no intersection. We randomly sample one image from the video clip as the visual message, which is resized to $256 \times 256$ then randomly cropped to $224 \times 224$. The audio messages are first re-sampled into 16K Hz, then translated into spectrogram via Short Time Fourier Transform with a Hann window length of 160 and a hop length of 80. Similarly with [31, 16], Log-Mel projection is performed over the spectrogram to better represent sound characteristics, which therefore becomes a $201 \times 64$ matrix. The audio and visual message from the same video clip are deemed as a matched pair, otherwise mismatched. We use variants of ResNet-18 [13] as audio and visual feature extractors. Detailed architecture is shown in the materials. Our model is trained with Adam optimizer with learning rate of $10^{-4}$. In training phase, we use a threshold of 0.05 to binarize the localization maps to obtain object mask, with which we can extract object representations over feature maps. And each center representation in the object dictionary is accordingly assigned to one object category, which is then used for class-aware localization evaluation. Note that, the proposed model is evaluated and trained on the identical dataset.

**Evaluation metric** We employ *Intersection over Union* (IoU) and *Area Under Curve* (AUC) as evaluation metrics for single source sound localization, which are calculated with predicted sounding area and annotated bounding box. For discriminative sounding object localization in cocktail-party, we introduce two new metrics, *Class-aware IoU* (CIoU) and *No-Sounding-Area* (NSA), for quantitative evaluation. CIoU is defined as the average over class-specific IoU score, and NSA is the average activation area on localization maps of silent categories where the activation is below threshold $\tau$,

$$CIoU = \frac{\sum_{k=1}^{K} \delta_k IoU_k}{\sum_{k=1}^{K} \delta_k}, \quad NSA = \frac{\sum_{k=1}^{K}(1-\delta_k)\sum s^k < \tau}{\sum_{k=1}^{K}(1-\delta_k)A}, \quad (9)$$

where $IoU_k$ is calculated based on the predicted sounding object area and annotated bounding box for the $k-$th class, $s^k$ is localization map of $k$-th class, $A$ is the total area of localization map. The

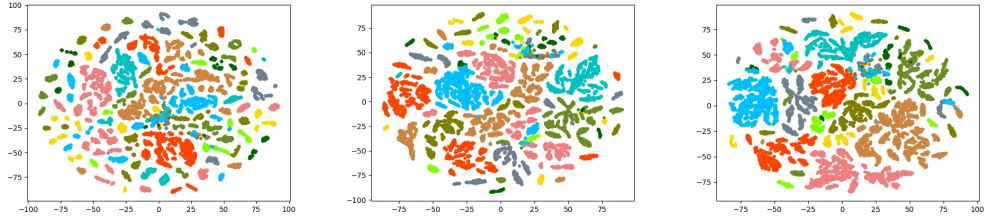

Figure 3: Visual feature distribution visualized by t-SNE on MUSIC-solo. These figures from left to right are global image features without alternative learning, image features with alternative learning and masked object features with alternative learning. The categories are indicated in different colors.

Table 2: Localization results on MUSIC-synthetic, MUSIC-duet and AudioSet-instrument-multi. Note that, the CIoU reported in this table is CIoU@0.3, and NSA of DMC is not evaluated since it relies on given knowledge to determine whether the object is sounding or silent.

| Data | MUSIC-Synthetic | | | MUSIC-Duet | | | AudioSet-multi | | |
|---|---|---|---|---|---|---|---|---|---|
| Methods | CIoU | AUC | NSA | CIoU | AUC | NSA | CIoU | AUC | NSA |
| Sound-of-pixel [31] | 8.1 | 11.8 | 97.2 | 16.8 | 16.8 | **92.0** | 39.8 | 27.3 | **88.8** |
| Object-that-sound [3] | 3.7 | 10.2 | 19.8 | 13.2 | 18.3 | 15.7 | 27.1 | 21.9 | 16.5 |
| Attention [27] | 6.4 | 12.3 | 77.9 | 21.5 | 19.4 | 54.6 | 29.9 | 23.5 | 4.5 |
| DMC [16] | 7.0 | 16.3 | - | 17.3 | 21.1 | - | 32.0 | 25.2 | - |
| Ours | **32.3** | **23.5** | **98.5** | **30.2** | **22.1** | 83.1 | **48.7** | **29.7** | 56.8 |

indicator $\delta_k = 1$ if object of class $k$ is making sound, otherwise 0. These two metrics measure the model's ability to discriminatively localize sounding objects and filter out the silent ones.

### 4.3 Single sounding object localization

In this subsection, we focus on the simple task of sound localization in single source scenario. Table 1 shows the results on MUSIC-solo and AudioSet-instrument-solo videos, where ours is compared with recent SOTA methods. Note that we use the public source code from [31, 16]. According to the shown results, we have two points should pay attention to. First, the compared methods [3, 16, 27] are trained to match the correct audiovisual pair via the contrastive [16, 27] or classification[3] objective, which is similar to ours. Yet, our proposed method significantly outperform these method by a large margin. Such phenomenon indicates that the learned object representations from localization is effective for semantic discrimination, which further benefits the object localization via the discriminative learning of object category. In order to explain this clearly, we plot the distribution of extracted feature from the well-trained vision network via t-SNE [19]. As shown in Fig. 3, the extracted visual features on MUSIC-solo are more discriminative in terms of object category when we train the model in a localization-classification alternative learning fashion, where the normalized mutual information for the clustering with masked object features achieves 0.74, which reveals high discrimination of learned representations. Second, our method is comparable to Sound-of-pixel [31], especially on the MUSIC-solo dataset. This is because Sound-of-pixel [31] differently employs the audio-based mix-then-separate learning strategy, which highly relies on the quality of input audio messages. Hence, it could effectively correlate specific visual area with audio embeddings in the simple scene with single sound, but suffers from the noisy multi-source scenarios. In contrast, our method can simultaneously deal with both conditions and does not require to construct complex learning objective. Related results can be found in the next subsection.

### 4.4 Multiple sounding objects localization

Natural audiovisual scenario usually consists of multiple sounding and silent objects, which is more challenging for exactly localizing the sounding ones. To responsibly compare different methods under such scenarios, both of the synthetic and realistic data are evaluated. As shown in Table 2, we can find that our model shows significant improvements over all the compared methods in terms of CIoU. Such phenomenon mainly comes from three reasons. First, our model takes consideration of the class information of sounding objects by employing a category-based audiovisual alignment, i.e., Eq. 7, while other methods [3, 27] simply correlate the audiovisual features for sounding area detection so that fail to discriminatively localize the sounding objects. Second, our localization results are achieved with the effective visual knowledge learned from the first-stage, which could vastly help to

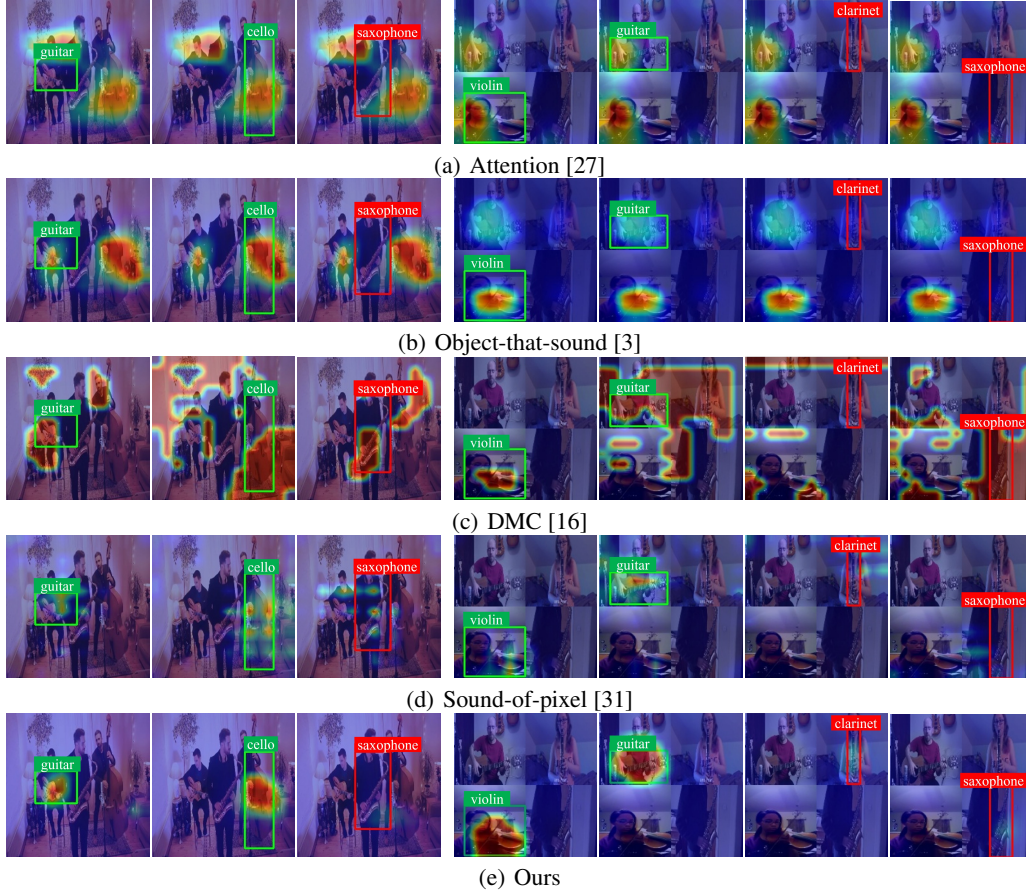

(a) Attention [27]

(b) Object-that-sound [3]

(c) DMC [16]

(d) Sound-of-pixel [31]

(e) Ours

Figure 4: We visualize some localization results of different methods on realistic and synthetic cocktail-party videos. The class-aware localization maps are expected to localize objects of different classes and filter out silent ones. The green box indicates target sounding object area, and the red box means this class of object is silent and its activation value should be low.

excavate and localize potential objects from cocktail-party scenario, while the compared method [31] cannot deal with such scenario with mix-then-separate learning fashion. Third, referring to NSA results, our model can automatically filter out the silent objects, but DMC [16] has to rely on given knowledge of the number of sounding objects. Although [31] is high in NSA, it is probably because of too low channel activations to detect objects rather than the success of filtering out silent ones.

Apart from the quantitative evaluation, we also provide visualized localization results in Fig. 4. According to the shown results in realistic scenario, the attention-based approach [27] and Object-the-sound [3] can just localize the sounding area without discriminating guitar or cello, while DMC [16] suffers from the complex audiovisual components and mix up different visual areas. Among these compared methods, although sound-of-pixel [31] provides better results, it cannot exact localize the sounding object and filter out the silent saxophone. This is probably because it highly depends on the quality of mixed sounds. In contrast, our model can successfully localize the sounding guitar and cello in class-specific maps, as well as remain low response for the silent saxophone and other visual areas. The synthetic data show similar results.

## 4.5 Ablation study

In this section, we perform ablation studies w.r.t. the influence of hyper-parameters. More studies can be found in the supplementary material.

**Loss function weight** $\lambda$. As shown in Table 3, we can find that the hyper-parameter of $\lambda$ has slight effects on the localization performance when in the range of $[0.5, 1.0]$. But it takes higher influence when becomes smaller or larger. Such phenomenon comes from the fact that the localization objective $\mathcal{L}_1$ is easier to converge compared with the distribution matching objective $\mathcal{L}_c$. When $\lambda$ becomes

Table 3: Influence of the hyper-parameter $\lambda$ on the MUSIC-Synthetic and duet dataset.

| Dataset | Music-Synthetic | | | Music-Duet | | |
|---|---|---|---|---|---|---|
| $\lambda$ | CIoU@0.3 | AUC | NSA | CIoU@0.3 | AUC | NSA |
| 0.3 | 11.0 | 15.0 | 94.6 | 33.7 | 23.9 | 81.7 |
| 0.5 | 32.3 | 23.5 | 98.5 | 30.2 | 22.1 | 83.1 |
| 0.8 | 27.9 | 21.4 | 96.2 | 26.5 | 21.7 | 79.9 |
| 1.0 | 28.6 | 21.1 | 96.5 | 33.2 | 22.9 | 83.1 |
| 1.5 | 13.2 | 15.4 | 94.1 | 24.5 | 20.0 | 84.6 |

Table 4: Ablation study on mask threshold and number of clusters.

| Dataset | | Music-Solo | | | Music-Synthetic | | | Music-Duet | | |
|---|---|---|---|---|---|---|---|---|---|---|
| Mask | Cluster | NMI | IoU | AUC | CIoU | AUC | NSA | CIoU | AUC | NSA |
| 0.05 | 11 | 0.748 | 51.4 | 43.6 | 32.3 | 23.5 | 98.5 | 30.2 | 22.1 | 83.1 |
| 0.03 | 11 | 0.752 | 49.2 | 45.3 | 31.4 | 24.2 | 96.2 | 29.3 | 22.0 | 81.7 |
| 0.07 | 11 | 0.753 | 43.1 | 40.7 | 32.1 | 24.0 | 94.5 | 29.7 | 21.6 | 80.3 |
| 0.05 | 13 | 0.735 | 47.6 | 43.7 | 33.8 | 24.0 | 96.2 | 30.3 | 21.9 | 83.2 |
| 0.05 | 20 | 0.720 | 44.6 | 45.4 | 29.5 | 22.2 | 98.9 | 28.9 | 20.1 | 84.2 |

much larger, the model would suffer from the overfitting problem for localization. When $\lambda$ becomes much smaller, it is difficult to achieve reasonable sounding area detection for effective filtering.

**Number of clusters and mask threshold.** In previous experiment settings, we set the number of clusters at the first stage equal to the number of categories in the dataset, which provides a strong prior. Therefore, we explore using different number of clusters as well as the mask threshold for the first-stage object feature extraction and clustering. And for evaluation, we adpatively aggregate multiple clusters to one specific category for discriminative localization. Table 4 shows the results on Music dataset. It is clear that our method is generally robust to these to hyper-parameters, and achieves comparable performance without knowing the specific number of categories in the dataset.

Table 5: Ablation study for the second stage.

| Dataset | | | Music-Synthetic | | | Music-Duet | | |
|---|---|---|---|---|---|---|---|---|
| $\mathcal{L}_1$ | Prod | $\mathcal{L}_c$ | CIoU | AUC | NSA | CIoU | AUC | NSA |
| ✗ | ✗ | ✔ | 0.0 | 7.2 | 91.0 | 20.8 | 15.9 | 78.0 |
| ✔ | ✗ | ✔ | 2.6 | 7.5 | 88.1 | 20.6 | 20.2 | 79.6 |
| ✔ | ✔ | ✗ | 18.0 | 17.4 | 92.9 | 22.4 | 19.2 | 85.0 |
| ✔ | ✔ | ✔ | 32.3 | 23.5 | 98.5 | 30.2 | 22.1 | 83.1 |

**Training settings for the second stage.** We further present some ablation studies on the procedure and training objective for the second stage. We denote the localization loss as $\mathcal{L}_1$, the audiovisual consistency loss as $\mathcal{L}_c$, and the silent area suppress operation as Prod. As shown in the Table 5, the product operation is crucial especially under the synthetic circumstance. It is because in our manually synthesized data, there are totally four instruments in a single frame, with two making sound and the other two silent. If without the Prod operation, all the objects would produce high response, thus making the categorical matching between audio and visual components fail and leading to very poor performance. On the other hand, the $L_c$ objective boosts localization on both synthetic and real-world duet data, which demonstrates that evacuating inner consistency between two modalities helps cross-modal modeling.

## 5 Discussion

In this paper, we propose to discriminatively localize sounding objects in the absence of object category annotations, where the object localization in single source videos are aggregated to build discriminative object representation and the audiovisual consistency is used as the self-supervision for category distribution alignment. Although the object semantic learned from simple cases contributes noticeable results, it still need rough partition of single and multiple source videos, which should be emphasized in the future study.

## Acknowledgement

This work was supported in part by the Beijing Outstanding Young Scientist Program NO. BJJWZYJH012019100020098 and Public Computing Cloud, Renmin University of China.

## Broader Impact

Visually sound source localization is a kind of basic perception ability for human, while this work encourages the machine to be equipped with similar ability, especially when faced with multi-source scenarios. Hence, the impact mainly lies in the machine learning technique and application aspect. On the one hand, the proposed approach is fully based on self-supervised learning, but can reward considerable discrimination ability for the visual objects and correlation capabilities across audio and visual modalities. Predictably, without elaborately manual annotation, this approach could still facilitate the progress of unimodal and multimodal learning and parse/model complex scene. On the other hand, it steps forward to pursuing human-like multimodal perception ability, which could further contribute to our society in several aspects, e.g., audio-assistant scene understanding for the deaf people by figuring out which objects are making sound, facilitating exploration into how to solve the cocktail-party effect in realistic audiovisual scenes, i.e., to perceive different sounds and focus on the pertinent content from mixed auditory input.

## Footnotes

\*Corresponding Author, Beijing Key Laboratory of Big Data Management and Analysis Methods, Gaoling School of Artificial Intelligence, Renmin University of China, Beijing 100872, China. The research reported in this paper was mainly conducted when the corresponding author worked at Baidu Research.

[2]The cosine similarity is followed by a parameterized sigmoid function to achieve comparable scale to the binary supervision. More details about similarity computation and networks are in the material.

[3]The audio network is trained with the pseudo label in the first stage, more details are in the materials.

[4]Available at `https://zenodo.org/record/4079386#.X4NPStozbb0`

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
