[Supplementary Material · material.pdf]

# A    Supplementary Material[1]

## A.1    Architecture Details

**BackBone.** We use a simple modification of ResNet-18[2] obtaining audio and image embeddings. We increase the feature resolution by removing stride from the first convolution in the last stage. We also remove *Global Average Pooling* (GAP) and the final classifier. The corresponding model is so called ResNet-18-S5 and this backbone is used among all the experiments.

**Audio Network.** In order to fit the one channel audio input (spectrogram), we change the first convolution channel in audio ResNet-18-S5 from 3 to 1. Besides, the final GAP is replaced by *Global Max Pooling* GMP for suppressing no-sounding area. This output is $g(a_i^s)$ mentioned in the main paper. For classification task, a FC-based classifier whose dimension is based on the number of instruments containing in the dataset is appended to $g(a_i^s)$.

**Image Network.** Image network uses standard ResNet-18-S5 model. And we do not load ImageNet pretrained weights. $f(v_i^s)$ is the output of the image network. Similarly to audio network, GAP and classifier are employed for the classification task.

Table 1: Specific Operations for the AV Localization. Conv(m), k×k means the output channel of the convolution is m and kernel size is k. FC(d) means the output dimension is d.

| A-Net Loc Embedding | Feature Dims | I-Net Loc Embedding | Feature Map Shapes |
|---|---|---|---|
| | 512 | | 512×14×14 |
| FC(128) | 128 | Conv(128), 1×1 | 128×14×14 |
| FC(128) | 128 | Conv(128), 1×1 | 128×14×14 |
| Unsqueeze | 128×1×1 | - | 128×14×14 |
| L2-norm | 128×1×1 | L2-norm | 128×14×14 |
| Broadcast | 128×14×14 | - | 128×14×14 |
| AV-Localization | | | |
| Operations | | Feature Map Shapes | |
| Cosine Similarity | | 1×14×14 | |
| Conv(1), 1×1 | | 1×14×14 | |
| Sigmoid | | 1×14×14 | |
| Global Max Pooling | | 1 | |

**AV Localization.** As shown in Table 1, before calculating av-location, we add fully connected layers or convolution layers for audio network and image network respectively, to adjust features for better localization. Then, we normalize two features using l2-norm to eliminate the impact of scale and broadcast the audio representation to match the spatial resolution of image network. After obtaining the cosine similarity (different from [1]) of two feature representations, we add a one channel convolution, sigmoid activation and GMP to get the final verification score. We supervise the localization objective learning by binary cross-entropy loss which means whether the audio-visual pairs are matched or not.

## A.2    Details in the Learning Phrase.

**Alternative Optimization in the First Stage.** We alternatively train classification and localization objective in the first stage. Pseudo-labels are obtained based on the K-means. Because the centering order of K-means is not kept in every iteration, we re-initialize classifiers in audio network and image network respectively at the beginning of the classification learning. When the classification accuracy is saturated or the Maximum number of iterations is reached, we stop the classification learning and switch to localization objective learning. The threshold $\tau$ for IoU/CIoU is determined by the 10% of the max value of the all the predicted localization map.

**Semantic Label Acquirement.** Since we train the whole model in a self-supervised fashion, the concrete semantic of object label is agnostic. In other words, we do not know the one-hot label $[1, 0, 0]$ means *Guitar* and $[0, 1, 0]$ means *Saxophone*. However, the learned object dictionary in

(a) Violin and Guitar         (b) Saxophone and Flute

(c) Guitar and Cello         (d) Flute and Accordion

Figure 1: We visualize some localization results of our method on realistic and synthetic cocktail-party videos. The class-aware localization maps are expected to localize objects of different classes and filter out silent ones. The green box indicates target sounding object area, and the red box means this class of object is silent and its activation value should be low.

Table 2: Pretrained model on MUSIC dataset.

| Dataset | Music-Solo | | | Music-Synthetic | | | Music-Duet | | |
|---|---|---|---|---|---|---|---|---|---|
| Pretrained | NMI | IoU | AUC | CIoU | AUC | NSA | CIoU | AUC | NSA |
| Scratch | 0.576 | 25.0 | 33.9 | 7.8 | 1.5 | 92.8 | 12.5 | 15.1 | 95.2 |
| ImageNet | 0.748 | 51.4 | 43.6 | 32.3 | 23.5 | 98.5 | 30.2 | 22.1 | 83.1 |

the first stage is category-aware, which just lacks the alignment between the concrete semantic and representation keys in the dictionary. Hence, we perform the alignment for better visualization and evaluation.

## A.3 Ablation study

In this section, we perform ablation studies w.r.t. the influence of training settings, and also evaluate the effects of alternating optimization.

**Pretrained model.** For experiments on MUSIC dataset, we use the ImageNet-pretrained ResNet-18 to initialize the visual backbone in the default settings while the audio backbone is trained from scratch. In this subsection, we explore the influence of training the visual backbone from scratch, and the results show that without the ImageNet-pretrained model initialization, the NMI on Music-Solo drops dramatically and the visual features are not discriminative enough for category-aware sounding object localization. Note that the low NSA metric can be attribute to the low correspondence on visual objects instead of the true ability to discriminate sounding and silent. It is probably because MUSIC dataset contains much fewer scenes than AudioSet-instrument. Therefore, the pretrained model is crucial to avoid overfitting on MUSIC, while the model on AudioSet-instrument performs well when trained from scratch.

**Alternative optimization.** To validate the effects of alternating optimization in the first-stage, we choose to perform the single sounding object localization without the alternating optimization between localization (Eq.1) and classification objective w.r.t pseudo-label, the built object dictionary is then served to the multi-source localization in the second stage. As shown in Fig.3, it is obvious that the localization results are significantly better when with alternating optimization. This is because the categorization task provides considerable reference for generalized object localization [3, 4], which further advances the quality of object mask in our task.

Table 3: Localization performance on the MUSIC-Synthetic dataset.

| Alternating opt. | CIoU@0.3 | AUC | NSA |
|:---:|:---:|:---:|:---:|
| w/ | 32.3 | 23.5 | 98.5 |
| w/o | 17.4 | 16.5 | 95.7 |

(a) Single Visual Object but with Multiple Sounds

(b) Album Cover

(c) Small Objects

Figure 2: Three kinds of failure cases.

## A.4 Failure Cases

There are three main failure cases in AudioSet-instrument-multi, (a) single visual object but with multiple sounds, (b) album cover and (c) small objects. For single object with multiple sounds, it is difficult to perform category-based audiovisual distribution matching of sounding objects. For album covers, since no objects appear in the image, no area should be activated, however some channels still have activations. For small objects, some sounding objects may not locate well due to the resolution of spatial feature maps.

## Footnotes

[1]The supplementary material for *Hu et al. Discriminative Sounding Objects Localization via Self-supervised Audiovisual Matching, In NeurIPS 2020*.