[Reviews · NeurIPS 2020]

Review 1

Summary and Contributions: The authors propose a two-stage framework to tackle the task of discriminatively localising sounding objects in the cocktail party scenario without manual annotations. Robust object representations are first learned in a single source scenario, before expanding to a multi-source scenario where sounding object localisation is formulated as a self-supervised audiovisual consistency problem, which is solved through object category (audio and visual) distribution matching.

Strengths: + Curriculum learning from a simple scenario with a single sound source to a complex scenario with multiple sounding sources, i.e. cocktail party scenario. + The proposed method addresses natural audiovisual scenarios, which consist of multiple sounding and silent objects (unlike some prior works which do not address silent objects). It is nice to match audio and visual object category distributions. + Ablations are performed (sup mat) to show the benefit of alternating between localisation and classification for the first stage (single source). + In the single source scenario, the proposed method achieves either better or comparable results to Sound of pixels [30] (top performing baseline out of several shown) on the MUSIC-solo and AudioSet-instrument-solo dataset splits. + In the multiple source scenario, the proposed method outperforms all baselines on MUSIC-Synthetic, MUSIC-Duet and AudioSet-Multi dataset splits for CiOU and AUC metrics (although not NSA).

Weaknesses: 1. Some more ablations would be nice. The authors could for example investigate the impact of removing for the second stage (multi source scenario) the sounding area (li stage) as well as sounding object location (si). 2. Some implementation details are missing. E.g. how long do the authors train the first stage vs the second stage? More details would be better for reproducibility.

Correctness: Yes the method seems correct and the experimental section well executed.

Clarity: The paper is relatively well written. The problem, proposed framework and experiments are clearly explained.

Relation to Prior Work: Comparison to prior work is fairly well discussed. Unlike previous work, this method uses an established dictionary of object representations to predict class-aware object localization maps. This work addresses mixed sound localisation whereas previous methods assume mainly single source scenes.

Reproducibility: Yes

Additional Feedback: For the video in sup mat, it would be more clear if the authors' method was compared to baselines on the same slide. Typos: L.190: two learning objectives Sound-of-pixel -> Sound of pixels Object-that-sound -> Objects that sound ########### POST REBUTTAL ################ I thank the authors for their feedback. I agree with some of the issues raised by the other reviewers: For example the requirement for knowing the number of sources in a video as well as the availability of videos with single sound source for the first part of the curriculum are legitimate concerns. It is also true that the music instruments setting is relatively simple setting, however there's been lots of previous work on this kind of data that so I think it's fair as a benchmark. Overall, I believe the work is good, novelty is sufficient and merits acceptance. I therefore keep my initial score and recommendation. A further comment: Could the authors give some more details on which datasets the models have been trained on for each experiment (e.g. are the models that are evaluated on MUSIC trained on MUSIC only, Audioset only or both?)


Review 2

Summary and Contributions: The paper proposes methods to localize objects producing sounds in a given audiovisual scene. This is done in an unsupervised setting where manual annotations are not available. The framework first tries to learn robust object representations and then uses audiovisual consistency to train the networks to localize the sounding objects.

Strengths: Localizing sounding objects in a given audiovisual scene is an interesting problem. The paper presents a novel approach for this problem which does not require manual semantic labeling and the training is largely self-supervised and relies on inherent audiovisual consistencies for training the models and the overall approach is nice. Comparison with several prior methods has been done to show the superiority of the proposed method.

Weaknesses: One major limitation of the work is that only music related objects and sounds are used. This does not provide a good idea of how well this method generalizes for everyday objects and sounds produced by them. It would have been nice if this paper had considered this more general condition in their datasets. There are few other concerns w.r.t how the method will generalize. Please look at the detailed comments below.

Correctness: Yes, the method and the empirical methodology seems correct.

Clarity: The paper is well written and clear. There are a few sentences here and there which I feel can be restructured. For example, the last line of paragraph 1.

Relation to Prior Work: Yes

Reproducibility: Yes

Additional Feedback: The paper presents a framework for localizing sounding objects in an audiovisual scene. Overall, I liked the paper. The proposed approach is neat and makes sense to the most extent. I have a few points of concern and I would like to see the author's responses on them. I would be happy to raise my overall score if the responses are satisfactory. -- Post rebuttal -- Most of my concerns have been addressed and I think the paper should be accepted. Score has been updated to reflect that. 1. The method relies on knowing which videos have a single source and which ones don’t. Where is this information coming from ? For the datasets used, this seems to be known apriori, implying some sort of manual labeling. Is that the case ? 2. Is the number of object categories, K in Eq 3, known apriori ? Is it equal to the number of instruments for each dataset ? Seems like a strong assumption that this information will be known, especially given the claims around self-supervised learning. What would do if this information is not known ? 3. The output s_i^k gives us the location of sounding objects for the k^th object category. GAP(s_i^k) averages this out across all locations, giving us a score estimating the probability of presence of the sounding object in the scene. Now given that in multisource conditions, two objects might be making the sound, why do the softmax over all object categories ? 4. I believe the authors were just following prior works but I am surprised why the “balanced” set of Audioset is used for test. Audioset comes with a “Eval” set for testing. 5. How is the threshold for binarization for mask set to 0.05 ? Is it a factor in performance ? 6. As mentioned one major limitation of this work is that it is empirically studied only in a music setting. I think this is a relatively easier condition and feels very synthetic compared to a more realistic one where different types of objects producing different types of sounds are considered. I understand some prior works have done the same. 7. I am glad that the authors commented on 3 failure cases. What about situations when objects are partially occluded ? How well do you think it will work ? --------------- While this paper does have limitations -- reliance on availability of solo videos, knowing number of sound sources in the dataset, scaling to general everyday objects, overall I think it is still a good paper. Several of my concerns have been addressed in the rebuttal and I have updated my score to reflect that.


Review 3

Summary and Contributions: This paper addresses the problem of sound source localization in videos frames, and the authors propose a two-stage approach that first learns representaions of sounding objects, and then perform class-aware object localization based on the learned object representations. Experiments demonstrate that the proposed approach leads to some accuracy gains for this task.

Strengths: Nice motivation to consider a two-stage framework that first learns object representation in single source scenario and then perform class-aware object localization maps in multi-source scenarios. Good results on sound source localiztion compared to prior methods in Table 2. Nice qualitative results on sound source localization.

Weaknesses: - It is claimed that the proposed method aims to discrminatively localize the sounding objects from their mixed sound without any manual annotations. However, the method aslo aims to do class-aware localization. As shown in Figure 4, the object categories are labeled for the localized regions for the proposed method. It is unclear to this reviewer whether the labels there are only for illustrative purposes? - Even the proposed method doesn't rely on any class labels, it needs the number of categories of potential sound sources in the data to build the object dictionary. - Though the performance of method is pretty good especially in Table 2, the novelty/contribution of the method is somewhat incremental. The main contribution of the work is a new network design drawing inspirations from prior work for the sound source localization task. - The method assumes single source videos are available to train in the first stage, which is also a strong assumption even though class labels are not used. Most in-the-wild videos are noisy and multi-source. It would be desired to have some analysis to show how robust the system is to noise in videos or how the system can learn without clean single source videos to build the object dictionary.

Correctness: The claims and the proposed method are correct as well as the empirical methodology.

Clarity: The paper is generally nice written and easy to follow.

Relation to Prior Work: The relations to prior work are well discussed and this works has major differences compared to previous contributions.

Reproducibility: Yes

Additional Feedback: - It would be useful to show what categories do the different colors represent. - the phrase audio/visual "message" sounds very strange #######################AFTER REBUTTAL#################### I concur with other reviewers on the merits of the paper, especially on the nice quantitave results and the design of the self-supervised training paradigm. The rebuttal has addressed most of my questions. However, I still have the following concerns: 1) Although the authors claim that the paper doesn't need any mannual annotions, they still need a dataset of solo videos (which doesn't come for free), and a rough estimate of the number of sources in the dataset. From this perspective, the Sound-of-Pixels baseline also doesn't need any mannual annotations. 2) The essential step of building an object dictionary makes the method hard to scale to more objects or more categories (e.g., general AudioSet videos instead of just instruments, as also pointed out by R2). Nevertheless, based on the merits of the paper, I would be fine to see this paper accepted if the authors can incorporate the proposed changes / additional results into camera ready, and make the limitations/distinctions clear.


Review 4

Summary and Contributions: This paper proposes to tackle sounding object localization in a cocktail party scenario, where the sounds are mixed and there might be silent objects. It also proposes a two-stage learning framework by first training an audiovisual localization network in single-sound scenarios and then using audiovisual consistency to match the distribution of visual objects and sounding objects.

Strengths: The proposed task is interesting and more realistic in real life. Their two-stage learning framework also has good quantitative results and beats other methods on most metrics.

Weaknesses: My biggest concern is that there is no quantitative ablation study on the effect of the audiovisual consistency objective in equation 7. Although the t-SNE plot shows alternative learning generates better visual features, there are no quantitative studies on how each stage affects the final results. And the lack of ablation makes the second contribution or technical contribution weaker because obviously, the novel part comes from using audiovisual consistency for category distribution matching. I find it interesting that related work including this work doesn't employ temporal information from the video for localization. For example, finger movement is one obvious visual clue of whether an instrument makes sounds.

Correctness: Yes. The claims are supported by experiment results.

Clarity: The writing is clear and easy to understand.

Relation to Prior Work: Yes. This paper has clearly discussed the difference with previous work.

Reproducibility: Yes

Additional Feedback:

[Author Response · NeurIPS 2020]

1 Many thanks for the precious comments. And we really appreciate the recognition to the contribution of this work!

**To Reviewer#1**

**Q1**: Some more ablations. **A1**: Table 1 shows the ablation study for the second stage on localization loss $\mathcal{L}_1$, sounding area suppress operation *Prod* and category matching objective $\mathcal{L}_c$. Our model achieves the best when with all of them.

**Q2**: How long for training the first stage vs the second stage? **A2**: Both two stages require about 10k training iterations to reach the best performance.

**To Reviewer#2**

**Q1**: The information about single/multiple sources. **A1**: The information comes from the dataset itself, e.g., solo/duet partition in MUSIC and number of annotated sound source in AudioSet. However, such strict partition does not always satisfy. We found that the AudioSet videos annotated as single source usually contain multiple sounding objects, e.g., *-g4i39nadkQ* and *MeGMI06BmLs* (YouTube video ID). Even faced with such noisy and incorrect partition, our method still generalizes well and provides good object representation for the second stage learning.

**Q2**: Number of object categories. **A2**: The number of object categories is equal to instruments in dataset for easy evaluation. However, Table 2 shows different cluster numbers, and our model still achieves comparable results.

**Q3**: Using softmax over all object categories? **A3**: 1. Using softmax can simultaneously encourage the model to find sounding objects and suppress the silent ones. 2. The followed KL divergence requires the input to be a distribution.

**Q4**: Using the balanced instead of the eval set in Audioset for testing. **A4**: For fairness, we just follow the previous works and use the balanced set for testing. As the balanced and eval set are constructed with same rules, the testing performance should be consistent. If permitted, we will append the comparison results on the eval set.

**Q5**: The influence of threshold for binarization for mask. **A5**: Table 2 shows the influence of mask threshold, 0.03, 0.05 and 0.07. Our method is robust to the choice of threshold.

**Q6**: Generalization on universal objects. **A6**: 1. Playing instruments is a typical audiovisual scenario, and generally used for evaluation previously [3, 30]. 2. We partially examine generalization on the challenging YouTube video (AudioSet). Although it also consists of playing instruments, these wild videos are very noisy, many are of poor quality and mixed with other object sounds [3]. 3. Although lack of benchmark dataset, we agree that the more general daily-object should be explored. In future, we will focus on these to further facilitate the development of sound localization.

**Q7**: Localization for occluded objects. **A7**: We show localization results for occluded objects of guitar and violin in Fig. 1, and we think our method is robust to partial occlusion as long as some key parts of objects are exposed.

**To Reviewer#3**

**Q1**: Whether the labels there are only for illustrative purposes? **A1**: The label in Fig.4 is only for illustrative purpose and not used in the training stage. The details about semantic label acquirement can be found in the sup material.

**Q2**: Number of object categories. **A2**: We explore using different number of clusters in training in Table 2, our method is robust to the cluster number. Hence, we do not certainly require it in training, but use it for evaluation.

**Q3**: The novelty/contribution. **A3**: 1. Unlike the simple audiovisual cases in previous works, this work deals with a more realistic and complicated cocktail-party scenario (confirmed by R#1 and R#4), meanwhile targets to discriminatively localize sources without manual semantic label. 2. The proposed techniques, including robust object representation learning, self-supervised category matching, step-by-step learning paradigm etc, are obviously different from previous localization approaches and recognized by R#1 and R#2.

**Q4**: Noisy single source video. **A4**: Please see A1 to R#2.

**To Reviewer#4**

**Q1**: Quantitative ablation study in Eq. 7. **A1**: In Table 1, without the consistency objective $\mathcal{L}_c$ but only scene-level correspondence, there is no supervision to facilitate discriminative localization, causing the dramatic performance drop.

**Q2**: Temporal information for localization. **A2**: Yes! Temporal information can be considered as a potential cue for sound localization, and we plan to explore this in future. Many thanks for the comment on such promising direction.

Figure 1: Occluded obj.

Table 1: Ablation study for the second stage.

| Dataset | | | Music-Synthetic | | |
|---|---|---|---|---|---|
| $\mathcal{L}_1$ | Prod | $\mathcal{L}_c$ | CIoU | AUC | NSA |
| ✗ | ✗ | ✔ | 0.0 | 7.2 | 91.0 |
| ✔ | ✗ | ✔ | 2.6 | 7.5 | 88.1 |
| ✔ | ✔ | ✗ | 18.0 | 17.4 | 92.9 |
| ✔ | ✔ | ✔ | 32.3 | 23.5 | 98.5 |

Table 2: Ablation on threshold, cluster number.

| Dataset | | Music-Synthetic | | |
|---|---|---|---|---|
| Threshold | Cluster | CIoU | AUC | NSA |
| 0.05 | 11 | 32.3 | 23.5 | 98.5 |
| 0.03 | 11 | 31.4 | 24.2 | 96.2 |
| 0.07 | 11 | 32.1 | 24.0 | 94.5 |
| 0.05 | 13 | 33.8 | 24.0 | 96.2 |
| 0.05 | 20 | 29.5 | 22.2 | 98.9 |

[Meta-Review · NeurIPS 2020]

All four reviewers reached a consensus that the paper passes the acceptance bar of NeurIPS. Despite its incremental nature, the proposed approach achieves strong results in challenging scenarios, compared to previous methods. The AC agrees with the recommendation made by the reviewers. However, as pointed out by R3, the claim that the method doesn’t need manual annotations is inaccurate (given the need for solo videos). The authors need to fix this issue, clearly articulate the limitations of the method, and add the discussion in the rebuttal to the final version of the paper.